# Determination of the Spatial Resolution in the Case of Imaging Magnetic Fields by Polarized Neutrons

Wolfgang Treimer [1,*] and Ralf Köhler [1,2]

1  Department of Mathematics, Physics & Chemistry, University of Applied Sciences,
   Berliner Hochschule für Technik Berlin, 13353 Berlin, Germany; ralf.koehler@drfz.de
2  Immune Dynamics, Deutsches Rheuma-Forschungszentrum (DRFZ), a Leibniz Institute,
   Charitéplatz 1, 10117 Berlin, Germany
*  Correspondence: treimer@beuth-hochschule.de

**Abstract:** One of the most important parameters characterizing imaging systems (neutrons, X-rays, etc.) is their spatial resolution. In magnetic field imaging, the spatial resolution depends on the (magnetic) resolution of the depolarization of spin-polarized neutrons. This should be realized by different methods, but they all have in common that a spin-polarizing and spin-analyzing system is part of the resolution function. First a simple and useful method for determining the spatial resolution for unpolarized neutrons is presented, and then methods in the case of imaging with polarized neutrons. Spatial resolution in the case of polarized neutron imaging is fundamentally different from 'classical' spatial resolution. Because of $\pi$-periodicity, the shortest path along which a spin-flip can occur is a measure of 'magnetic' spatial resolution. Conversely, the largest detectable magnetic field (B-field) within a given path length is also a measure of magnetic spatial resolution. This refers to the spatial resolution in the flight direction of the neutrons ($\Delta y$). The $\Delta x$ and $\Delta z$ refers to the spatial resolution in x- or z-direction; however, in these cases a different method must be used. Therefore, two independent methods are used to distinguish longitudinal and lateral spatial resolution, one method to determine the modulation transfer function (MTF) by recording the frequency-dependent fringe contrast of magnetic field images of a coil (longitudinal spatial resolution), and the second method, to observe the fringe displacement at the detector as a function of magnetic motion, provided that the accuracy of the motion is much better than the pixel size (at least half the pixel size) of the detector (lateral spatial resolution). The second method is a convolution of the fringe pattern with the pixel array of the detector.

**Keywords:** neutron imaging; polarized neutrons; magnetic fields; spatial resolution





## 1. Introduction

To recognize and distinguish small structures, inhomogeneities, defects, etc. in matter, it is necessary to know some basic conditions in neutron imaging regarding spatial resolution, beam divergence, contrast, etc. The theory to this subject is well established and described in detail in [1–5]. The spatial resolution of a neutron imaging system can be measured by several methods, two of them are frequently used and commonly accepted. The first one uses a so-called "Siemens star" and the second method by determining the modulation transfer function (MTF) of the imaging system. The determination with the "Siemens star" is simple because one can directly read the resolution from an image of the Siemens star. The disadvantage is that it does not cover the frequency range continuously.

Other methods use (absorbing) grids with different line pairs (LP) per unit length and plot the image contrast as a function of LP/unit length or calculate the MTF from the image of an edge. The more common method uses opaque edges, because absorbing grids with different line pairs (LP) per unit length yield only a discrete Fourier spectrum, different to an absorbing edge. There is a large number of publications showing up all advantages and disadvantages of this method, but using an edge is a simple but very straightforward

method to derive the modulation transfer function of a digital radiography system from an edge image. The image examined with the point spread function (PSF) method must have a high contrast-to-noise ratio, so it is (for clinical studies) often inappropriate for describing the MTF as well as with the edge spread function (ESF) method [6,7]. A technique for rendering the ESF method robust to image noise is presented where the noisy ESF is smoothed out through multiple stages of filtering [8]. The MTF can be also measured directly as the image contrast as a function of the spatial frequency of absorbing gratings with increasing number of line pairs/unit length [9].

The spatial detector resolution approaches the 1 μm size, driven by different techniques such as micro-channel plate (MCP), microscope-like devices, gadolinium oxysulfide scintillators, absorbing grids, observing tracks from neutron capture events, hybrid pixel detector (Medipix [10]), coded source imaging (CSI) system [11], fiber optics taper (FOT), etc. [8,12–21], and inaccuracies associated with use of precision test objects, such as a slit or an edge can be reduced [22,23] and misalignment of an edge with respect to detector pixel columns or rows can be taken into account [6]. A collection of test devices was developed for neutron imaging that can be used to quantify pixel and voxel size, resolution of the imaging system, and beam divergence. [24]. In 2014 the spatial resolution in neutron imaging reached with 7.8 μm = 63.2 line pairs/mm a level, which is comparable with X-ray imaging, [25]; however, today already a resolution of X-ray imaging is obtained [15,20,21,26–28].

For radiography (and tomography) using polarized neutrons the magnetic spatial resolution is always associated with change of the neutron spin orientation due to a magnetic field. Polarized neutron imaging is used in many experiments and has been published, e.g., [29–40], but all have in common a spin analysis system which determines the resolution of the imaging system.

In this paper, first a simple and powerful method is proposed to determine the spatial resolution in the case of imaging with un-polarized neutrons. Then we describe how to determine the spatial resolution in the case of imaging with polarized neutrons. We use the image of de-polarized neutrons by special magnetic fields that produces fringe structures with different frequencies in the space domain and additionally the shift of a given pattern with respect to detector pixel array. However, first, some mathematical basics about conventional spatial resolution are summarized.

## 2. Some Mathematics on Spatial Resolution

### 2.1. Spatial Resolution for Unpolarized Neutrons

Spatial resolution principally depends on the incident beam divergence $\phi$, in neutron radiography and tomography it is called L/D ratio, the inverse number of $\sim \phi$. There are different definitions of the L/D ratio (see [4]); however, here we use the most common accepted one, depicted in Figure 1. Conventional neutron imaging instruments use a pinhole aperture (size D) as source, which together with the distance L from the object determines the divergence $\phi$ of the incident beam. Depending on the size of the source D, or on the mosaic spread of a crystal source $D_{mosaic}$ a point in a sample is magnified on a screen as d = $l_d$D/L (Figure 1).

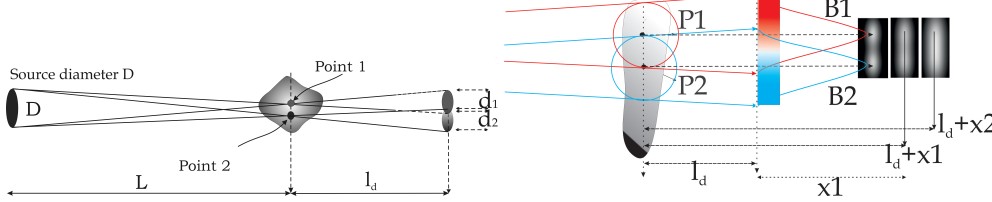

**Figure 1.** To distinguish two points in a sample, L, D, $l_d$ must be matched to each other.

In the case of a crystal monochromator, D must be replaced by $d_{mosaic}$, $d_{moasic}$ = mean grain size of the crystal monochromator and L/D then becomes $L/d_{mosaic}$ [41]. One can

pay attention to the refraction of the beam at edges that causes additional tiny blurring in the image, but refraction is indeed used as well as an imaging signal. This is not to be confused with the additional blur caused by refraction. Here, edge diffraction can be neglected, details on this subject are given in [42–47].

A finite L/D ratio is always linked with beam divergence $\phi$ and $\Delta\lambda/\lambda > 0$. Furthermore, a finite precision of scanning and detection (pixel size) contribute as well to smearing effects. To briefly recapitulate the mathematics of spatial resolution, divergence and blur, one considers a perfect straight edge, which is represented by the Heaviside step function $h(x)$. If an edge is illuminated by divergent radiation, given by $r = r(x)$, the geometrical image (shadow) is the convolution of $h(x) \otimes r(x)$. The function $r(x)$ includes divergence and all other blurring effects and can be best represented by a Gauß function $g(x)$,

$$g(x) = \exp\left(-\left[\frac{x - x_0}{2 \cdot \sigma}\right]^2\right) \tag{1}$$

The convolution $h(x) \otimes g(x)$ is given as

$$h(x) \otimes g(x) = \int_{-\infty}^{\infty} \left(\frac{1}{2} + \frac{1}{2} \cdot \text{sgn}(x)\right) \exp\left(-\left[\frac{(x - x') - x_0}{2 \cdot \sigma}\right]^2\right) dx' \tag{2}$$

The question to be answered is the "correct" scanning (sampling) of the image of the edge, which requires that the sampling frequency $f_s$ must be twice as high as the highest occurring frequency $f_{max}$ in the corresponding frequency spectrum, i.e., $f_s = 2\,f_{max}$ (Shannon theorem [48]). The scanning frequency is usually the number of detector pixels per unit length.

The highest occurring frequency $f_{max}$ in the corresponding frequency spectrum is obtained from the Fourier Transform $\mathcal{FT}$ of the convolution $h(x) \otimes g(x)$, C(f) = $\mathcal{FT}\{h(x) \otimes g(x)\}$. If $H(f)$ and $G(f)$ are the Fourier Transforms of $h(x)$ and $g(x)$ and using the convolution theorem

$$\mathcal{FT}\{h(x) \otimes g(x)\} = H(f) \cdot G(f), \quad H(f) = \mathcal{FT}\{h(x)\}, \quad G(f) = \mathcal{FT}\{g(x)\} \tag{3}$$

and with

$$\mathcal{FT}\{h(x)\} = \frac{1}{\pi} \cdot \delta(f) - \frac{1i}{2\pi f},$$

$$\mathcal{FT}\{g(x)\} = \sqrt{2 \cdot \pi} \cdot \sigma \cdot \exp\left(-\frac{f \cdot (f \cdot \sigma^2 + f_{k\cdot} \cdot 2i)}{2}\right) \tag{4}$$

one obtains the Fourier transform of the convolution $H(f) \cdot G(f)$. Please note that $H(f) \simeq \delta(f)$ function. The normalized absolute value of $\|G(f)\|$ yields the modulation transfer function (MTF) or frequency contrast function $C(f)$. The frequency at 10% contrast gives the spatial resolution in the image. Please note that vice versa the integration of the Gauß function yields the error function erf(x) which perfectly can fit to an image of an edge .

$$\int G(x)dx = \int e^{-((x-x_0)/\sigma)^2} dx = \text{erf(x)} \tag{5}$$

Some further remarks: Any measurement of the spatial resolution of an instrument or experiment must clearly state how far the sample is from the detector, its thickness/size and what 'source' is used (pinhole, monochromator), and what collimation. If the distance of the sample to the detector is large compared to its thickness/size (extension in neutron flight direction), one can assume a uniform spatial resolution. In the other case, the largest distance should be used to calculate the spatial resolution.

### 2.2. Longitudinal Spatial Resolution of a Magnetic Field

Regarding the spatial resolution dx, dy and dz in x-, y- and z-direction one can assume dx = dy = dz if no asymmetry of the imaging system is evident. The detection of the size of a magnetic field or to determine the magnitude of a magnetic field in a polarized neutron imaging depend on the number of spin rotations in the field. This measurement is based on the interaction of the neutron spin with B and provides information from B in only one direction, e.g., in the y-direction (neutron flight direction) in a $\{x, y, z\}$ coordinate system. The neutron spin in a magnetic field B behaves like a classical particle, since the neutron velocity $v \ll c$ and its spin vector $\vec{s}$ rotates (classically seen) around $\vec{B}$ with the Larmor frequency $\omega_l = \gamma_L \cdot B$, where $\gamma_L$ is the gyromagnetic ratio of the neutron ($\gamma_L = -1.83247 \times 10^8$ rad· s$^{-1}$·T$^{-1}$). From $\omega_L = \gamma_L \cdot B$ one obtains the rotation angle $\phi$, if t is the transit time of the neutron through the magnetic field B. With $\lambda = h/p$, $p = m_n \cdot v$ and the neutron mass $m_n = 1.67493 \times 10^{-27}$ kg [49] , $\phi$ is

$$\phi = \omega_L \cdot t = \gamma_L \cdot B \cdot t \quad \rightarrow \quad \phi = \frac{\gamma_L}{v} \int_{path} B \cdot ds = \frac{\gamma_L \cdot m}{h} B \cdot s \cdot \lambda \tag{6}$$

One realizes, only $|\vec{B}|$ = B is important, not the orientation of $\vec{s}$ with respect to $\vec{B}$, i.e., $\omega_L$ is independent of the angle between $\vec{s}$ and $\vec{B}$. The factor $\frac{\gamma_L \cdot m}{h}$ is constant = $-4.6321 \times 10^{14}$ [T$^{-1}$ m$^{-2}$], so a spin rotation $\phi$ around 180° (spin-flip) for a neutron wavelength $\lambda = 0.32$ nm = 3.2 and B = 1 mT would occur for a path length ~21.2 mm. From Equation (6), $\phi = \phi(B, s, \lambda)$ is a function of magnetic field, path length s and wavelength $\lambda$. The total differential of $\phi$, is

$$\Delta\phi = \frac{\partial\phi}{\partial B} \cdot \Delta B + \frac{\partial\phi}{\partial \lambda} \cdot \Delta\lambda + \frac{\partial\phi}{\partial s} \cdot \Delta s \tag{7}$$

and yields only a dependence on the path length 's', because in the sample B is constant, thus $\frac{\partial\phi}{\partial B} \cdot \Delta B = 0$, and the term $\frac{\partial\phi}{\partial \lambda} \cdot \Delta\lambda$ can be considered to be small (~0.06 rad for B = 4 mT, s = 10 mm, $\lambda$ = 0.32 nm and $\Delta\lambda$ = 1% see Figure 2). Thus, the longitudinal spatial resolution of a B field is coupled to the magnitude of B, to the wavelength and wavelength spread $\Delta\lambda/\lambda$ used. Figure 2 shows the path length dependence of a spin-flip on magnetic field for three wavelengths. One sees, the larger $\lambda$ the smaller the path length for a spin-flip.

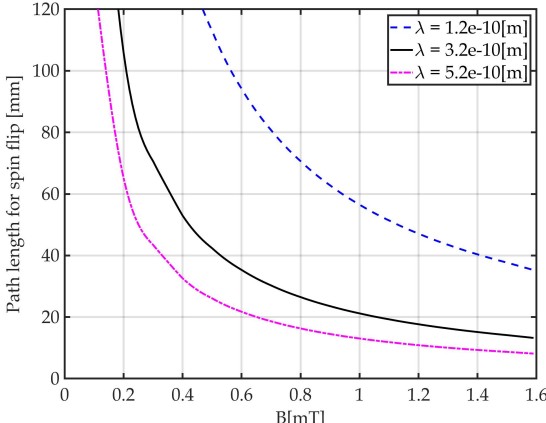

**Figure 2.** Dependence of path length for a spin flip as a function of magnetic field strength for three different wavelengths. If a neutron moves slower through B (large $\lambda$), the larger will be the spin rotation angle $\phi$ and therefore the shorter is the path for a spin-flip.

Therefore, a thin sample containing a magnetic field preferably requires a longer wavelength for the detection and determination of B and allows for a large $\Delta\lambda/\lambda \approx 10\%$ (thin in the sense of spin flips/sample thickness $\leq$ 6 spin flips, see Figure 3). $\Delta\lambda/\lambda$ determines the

contrast in polarized neutron images and one sees in Figure 3 that with increasing B the contrast of spin flips decreases. However it can happen, for the determination of B, that for a given wavelength different B-fields produce nearly the same gray values in an image, as shown in the 90° sample orientation in Figure 4, where a polarized neutron image of a coil shows for two different B-fields nearly the same gray values. On the other hand for $\lambda$ = 0.32 nm the difference are 5.5 gray values on a gray scale of 0–127, a small change of the wavelength to $\lambda$ = 0.34 nm increases the difference to 32 gray values as shown in Figure 5. The longitudinal spatial magnetic resolution $\Delta y$ is mainly a function of the path length s, s = s($s_{spin-flip}$), the size of B in the sample, on $\lambda$ and the wavelength spread $\frac{\Delta\lambda}{\lambda}$.

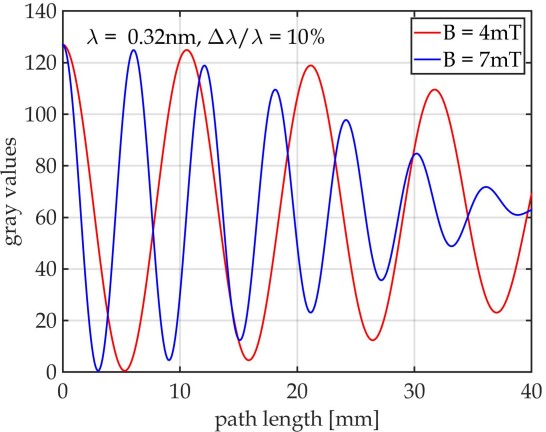

**Figure 3.** Contrast reduction of spin flips (minima and maxima in plot) due to a wavelength spread $\pm\Delta\lambda/\lambda \sim 10\%$ (mean $\lambda$ = 0.32 nm), red and blue line corresponds to magnetic fields B = 4 mT and 7 mT. The larger $\Delta\lambda/\lambda$ and B the shorter the path length for a number (6) of spin flips.

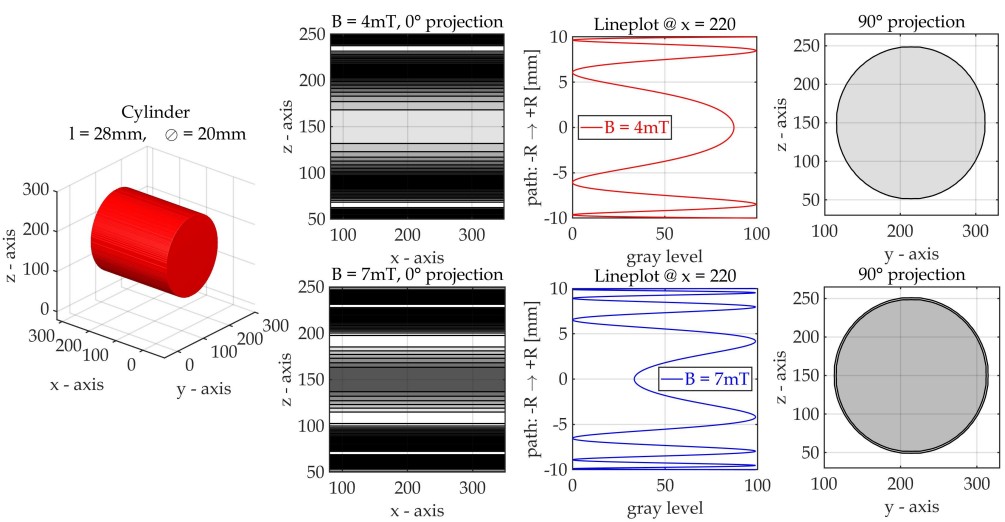

**Figure 4.** Calculated polarized neutron images: magnetized coil (red) with B-fields = 4 mT and 7 mT, four image projections due to polarized neutron imaging ($\lambda = 0.32 \times 10^{-9}$ m) for sample orientations 0° (2) and 90° (2) with respect to neutron flight direction = y-axis, and the corresponding line plots illustrating the effect of position-dependent $\int B \cdot ds$. Please note that the B-field is homogeneously distributed in the coil and therefore produces a homogeneous image at the 90° projections. The 4 mT B-field produces 5.3 spin flips, the 7 mT B-field 9.2 spin flips, so the gray values for the 90° orientations are very similar, gray scale = 0–127.

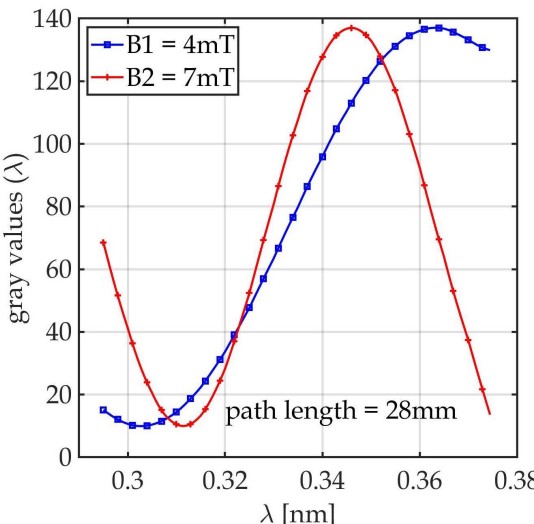

**Figure 5.** Two magnetic fields (4 mT and 7 mT) show nearly the same gray value for $\lambda$ = 0.32 nm and a path length = 28 mm, a small change of $\lambda$ = 0.34 nm increases the difference from 5 to 32 gray values, i.e., if the B-field in a sample is unknown, changing the wavelength may help identify B.

A longitudinal magnetic transfer function can be achieved by measuring the image fringe contrast as a function of different B with decreasing path length s. A fringe pattern is obtained by a proper coil operated with different currents, and measured by polarized neutron imaging (see Figure 6). The longitudinal spatial B resolution refers to the path length in the direction of the neutron trajectory through B. A fraction of this path length correlates with a fraction of the spin-flip and results in a different gray value in the image than a full spin-flip.

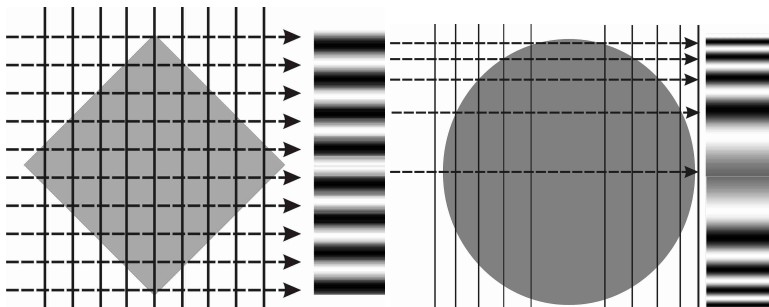

**Figure 6.** Calculated polarized neutron images: different fringe patterns due to different $\int B.ds$ in a coil, left: fringe pattern with a square coil, right: fringe pattern with a cylindrical coil.

For the $\{x, z\}$-plane (and any other plane), one obtains a frequency-dependent contrast function of different fringe patterns with a coil and imaged with polarized neutrons. The magnetic field in a coil of finite length L and radius R, B is given by [50]

$$B(x) = B_0 \cdot \left( \frac{x + \frac{L}{2}}{\sqrt{R^2 + \left(x + \frac{L}{2}\right)^2}} - \frac{x - \frac{L}{2}}{\sqrt{R^2 + \left(x - \frac{L}{2}\right)^2}} \right); \quad B_0 = \frac{\mu_0 \cdot n \cdot I}{2} \qquad (8)$$

where the axis of the coil is parallel to $\vec{B} = [B(x), 0, 0]$. For $L \gg R$ one obtains for $B(0) \approx \mu_0 \cdot n \cdot I$, and $B(\pm L/2) \approx 1/2 \cdot B(0)$ (see Figure 7).

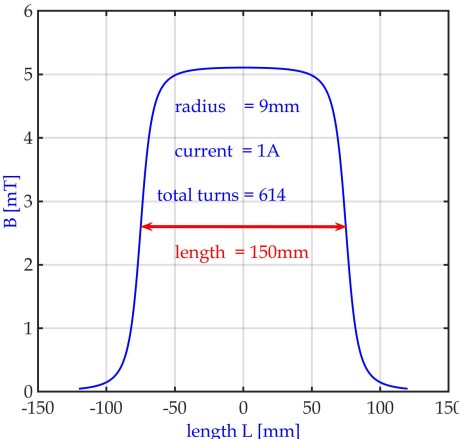

**Figure 7.** Calculated magnetic field in a coil, length = 150 mm, radius = 9 mm, windings = 614, current = 1 Ampere, B = 5.1 mT inside the coil.

The B-field in a coil with a length L much greater than the radius R, is known to be $\mu_0 \cdot N \cdot I / L$, $\mu_0 = 4\pi \cdot 10^{-7}$, N = number of winding, *I* = current and *L* = length. Figure 4 shows the calculated 2D polarized neutron images of a coil due to position-dependent depolarization of a polarized neutron beam, using the Radon Transform of the B-field of the sample. Such calculations (using the Radon Transform of a B-field) give perfect agreement with experimental results [36], even for rather complicated samples [51]. The images in Figure 4 were calculated with $B = \mu_0 \cdot N \cdot I / L$. For a coil, $\mu_o$, N and L are given device parameters, the current I and thus also B can be varied.

### 2.3. Lateral Spatial Resolution of a Magnetic Field

To acquire information about the lateral resolution one must use a different method. The lateral spatial resolution relates to the spatial symmetry mentioned above (Section 2.2). The divergence of B is zero, $\nabla \vec{B} = 0$, so B is constant in any volume element $\Delta V = \Delta x \times \Delta y \times \Delta z$, ($\Delta y \cong \Delta s$ in Equation (7)), one can further assume that the spatial resolution is $\Delta x = \Delta z$, i.e., if the horizontal and vertical divergence ($\phi_{hor}, \phi_{vert}$) are equal, otherwise, $\Delta z \neq \Delta x$ because the vertical divergence is $\phi_{vert} \neq \phi_{hor}$. The lateral spatial resolution is directly measured from moving a defined fringe pattern across the detector array, i.e., one measures the convolution of the pattern with the detector array. If $p = p(x, z)$ is the fringe pattern function and $D = D(x, z)$ the detector function, and one moves the pattern across the detector D, one measures the $I = I(x, z)$,

$$I(x, z) = \int_{-\infty}^{\infty} D(x, z') \cdot p(x, z - z') \cdot dz' \tag{9}$$

where *x* means one detector row. The convolution is described experimentally below (Section 3.3). Other considerations given Section 2.2 can be adopted for lateral magnetic resolution, too.

### 3. Experiments

### 3.1. Determination of the Spatial Resolution for Unpolarized Neutrons

The following experiments were performed at PONTO II, an instrument of the University of Applied Sciences, Beuth Hochschule Berlin, dedicated to polarized neutron imaging at the BER II reactor of the Helmholtz-Zentrum für Materialien und Energie. The first experiments with edge measurements were performed to determine the MTF of the system and thus the spatial resolution for unpolarized neutron imaging. A Cd edge (Figure 8) was placed in a cold neutron beam ($\lambda = 0.32$ nm) and an image taken at a distance of 60 mm from the 2D detector, which had a pixel size = 13.5 µm (Figure 8). From the fitted shape

of the smeared grid e.g., by a Gauß function, the Fourier Transform of the Gauß function, $\mathcal{FT}\{G\}$, yields the modulation transfer function MTF, the frequency-dependent contrast function C(f) of the edge image.

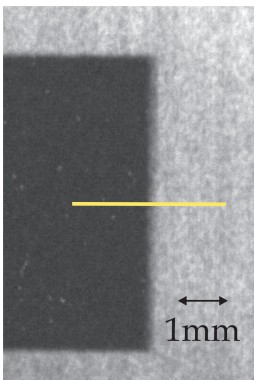

**Figure 8.** Image of a Cd edge, thickness 0.5 mm, neutron wavelength $\lambda = 0.32$ nm, exposure time = 1800 s, beam divergence $\phi = 0.1° \sim$ L/D = 573, [52].

Figure 9 shows steps (a)–(d) to obtain the MTF of the edge image and thus the spatial resolution of the instrument (see also Section 2.1). First the theory (a) is fitted to the experimental edge data (b), then this fit is adjusted to the error function by varying the FWHM of the Gauß function (c). This works better than using the error function itself, and above this, the Gauß function is fitted at the same time. The Fourier transform of the (fitted) Gauß function is then used to calculate the MTF = $(\mathcal{FT}(G(x))$ of the edge profile and to determine the 10% contrast $f_{max}$ and thus the spatial resolution of the instrument. The correct scanning frequency $f_{scan}$ is given by $f_{scan} = 2 \cdot f_{max}$. The detector pixel size was 13.5 μm but due to binning = 27 μm. The best spatial resolution is found at 54 μm ($\sim f_{max}$), which agrees perfectly with 2 × 27 μm ($\sim f_{scan}$) for $l_d$ = 5 mm. The spatial resolution also depends on the distance $l_d$ of the object to the detector (see Figure 1). The larger $l_d$ the worse is the spatial resolution. This was measured for $l_d$ = 5, 60, 100, 120, 140 and 200 [mm]. If one plots the particular spatial resolution as a function of $l_d$ one obtains the principal limit of the imaging system of $\sim$13 μm for $l_d = 0$, i.e., the pixel size of the detector 13.5 μm, which agrees perfectly with the data fit line for $l_d = 0$ mm (see Figure 10). Comparing the results in Figure 10 one sees the discrepancy between theory based on the $1/r^2$ law if a pin hole source is used and measured data. The measured data show an entirely different behavior. The explanation is given in [41], there already could be measured that using a mosaic crystal as monochromator the pin hole as source diameter must be replaced by the mosaic structure of the monochromator yielding an improvement of the L/D ratio of a factor $\sim$2.7, here a factor 2.2 (Figure 10).

The spatial resolution can be well derived by plotting the value of the modulation transfer function MTF at 10% contrast as function of the distance of the edge from the detector. For pin hole geometry on observes a $1/r^2$ behavior, any other geometry, such as using a mosaic crystal as source or a collimator system yields different results (see Figure 10).

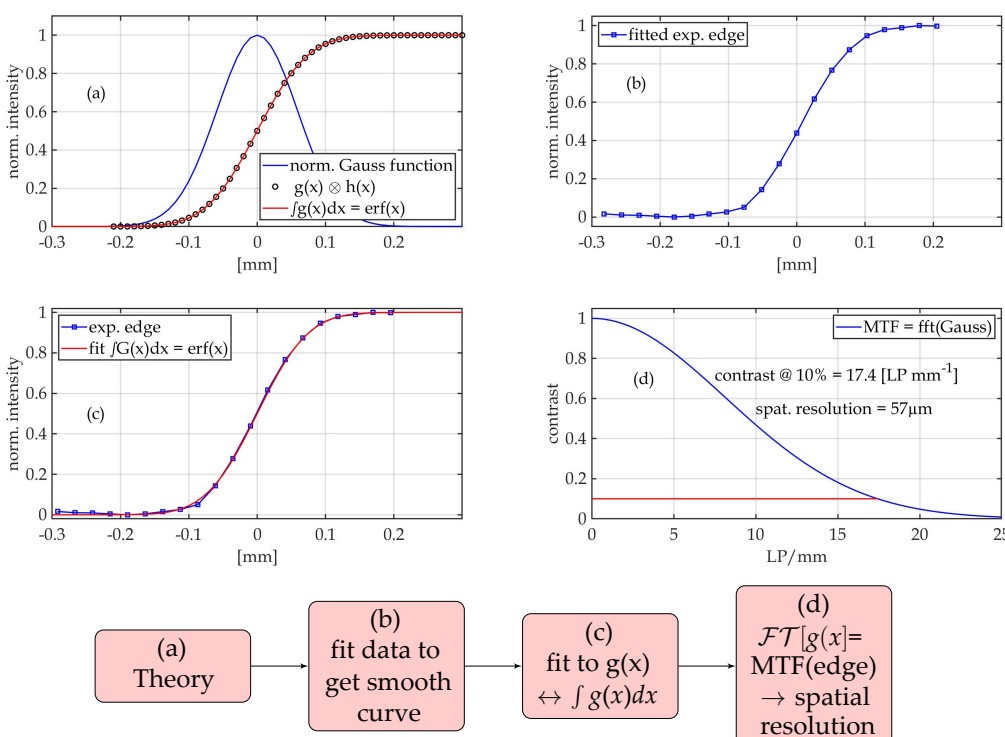

**Figure 9.** Fitting procedure and evaluation of the spatial resolution. The spatial resolution can be directly read from the 10% contrast at 17.4 LP/mm (appr. 57 μm). 17.4 LP/mm are the maximum frequency $f_{max}$ that appears in the image with 10% contrast.

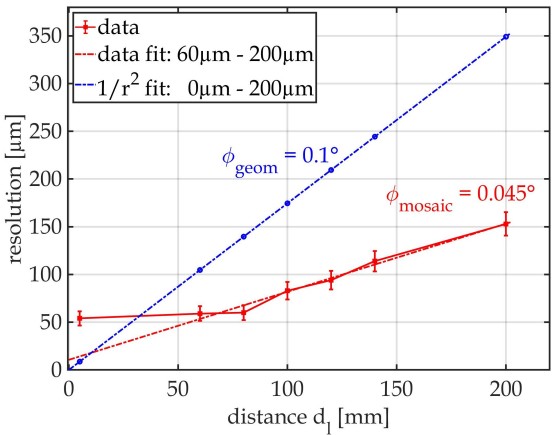

**Figure 10.** Spatial resolution as a function of distance of the edge from detector $l_d$. Note the difference between the measured and calculated resolution, the latter using the $1/r^2$ law and when the source geometry is assumed to be a pinhole rather than a mosaic crystal. $\phi_{geom} = 0.1°$ means the geometric collimation obtained with a solid-state collimator and $\phi_{mosaic} = 0.045°$ the measured beam collimation due to the mosaic crystal monochromator, see also [41].

### 3.2. Longitudinal Magnetic Spatial Resolution

In polarized neutron imaging, the magnetic modulation transfer function (mMTF) is the contrast function that depends on the number of neutron spin flips per path length in a magnetic field. It was determined as frequency-dependent fringe contrast function of neutron spin-depolarizing magnetic fields from a magnetized coil. The neutrons were spin-polarized and spin-analyzed by so-called solid-state benders (m = 2.5 and m = 3), length = 40 mm, cross section 30 mm × 30 mm [53], the neutron detector was 4k × 4k CCD camera. The fringe function as a 2D image of a pattern was generated by a magnetized coil

as calculated with Equation (6) and with the Radon Transform of the magnetic field [40]. Figure 6 shows the fringe pattern of coil with a quadratic and a cylindrical shape. A quadratic shaped coil creates equidistant fringe patterns different to a cylindrical shaped one and the fringe pattern is much easier to measure.

For the measurements we used a quadratic shaped coil (Al body 11 mm × 11 mm × 150 mm, width × height × length), Copper wire ($\oslash = 0.90(1)$ mm, resistance = 1.2 Ω) and currents up to 6 Ampere. The coil was placed 100 mm in front of the spin analyzer, which itself was directly in front of the 2D detector device (a CCD camera, 2k × 2k, pixel size 13.5 μm × 13.5 μm). The neutron beam ($\lambda$ = 0.32 nm) was collimated ($\phi_{hor} = 0.1°$, $\phi_{vert} = 0.2°$, $\phi_{hor}, \phi_{vert}$ are the horizontal and vertical divergences) and polarized and analyzed with Swiss Neutronics supermirror benders (m = 2.5 and m = 3, each 40 mm × 40 mm × 40 mm, the polarization P($\lambda$ = 0.32 nm) was > 96%, waver thicknesses of both supermirrors were 110 μm.

The spatial resolution was therefore limited to 110 μm in the $\{x, y\}$ plane and concerning the vertical divergence ($\phi_{vert}$ = 0.2°) in the $\{x, z\}$ plane, respectively (compare Figure 4). This geometry involved a vertical geometric smearing of 0.340 mm.

For each current I[A] a radiography with polarized neutrons was recorded as seen in Figure 11, exposure time was 5400 s. From each reduced image (size = 566 × 296 pixel $\cong$ 15.28 mm × 7.99 mm) the background was subtracted, and outlier data removed by a [5 × 5] median filter. Then a 150 pixel ( 4.05 mm) broad and 296 pixel (7.99 mm) high area was selected from the image and the contrast $C = (I_{max} - I_{min})/(I_{max} + I_{min})$ calculated as a function of fringes/7.99 mm. Figure 12 shows the result. The MTF function was fitted to an error function, yielding a vertical spatial resolution of 343 μm, which agrees well with the theoretical value of 0.340 mm Figure 12).

The fringe pattern at current I = 6A in Figure 11 is shown enlarged in Figure 13 which is under-laid with the corresponding line graph. It can be seen that the contrast of the fringes within the plot varies, mainly caused by small stray fields and imperfections of the neutron polarizer and the neutron analyzer, but the fringes are still well separated. The magnetic modulation transfer function was obtained by fitting the contrast values from the I = 3A, 3.5A ... 6A images with an error function and plotting the corresponding contrast as function of the frequency (Figure 12. Despite the rather small number of points (due to a limited coil current), a quite reliable value of 2.9 line pairs/8 mm at 10% contrast was obtained, corresponding to a resolution of 345(20) μm. The error can be derived from the different number of observed fringes (±1).

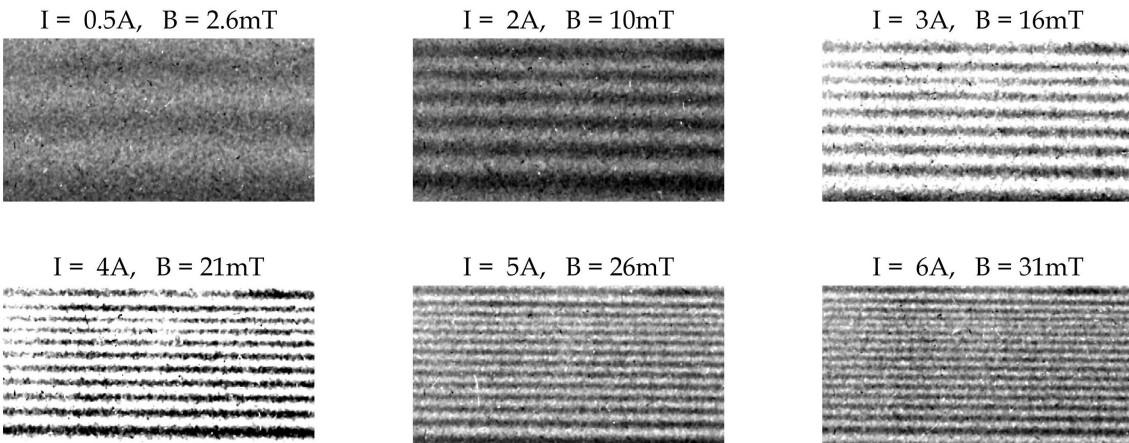

**Figure 11.** Radiography images of a magnetized coil due to path dependent depolarization of polarized neutrons. Fringe pattern occurs due to different path length in the quadratic coil, size = 566 × 296 pixel = 15.28 mm × 7.99 mm, exposure time = 5400 s, B = magnetic field in the coil, see fringe pattern of Figure 6.

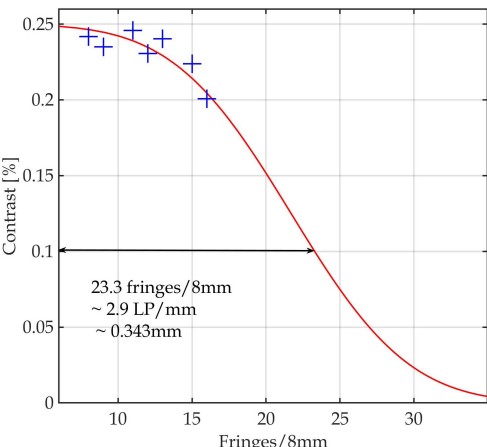

**Figure 12.** The "magnetic MTF" shows 2.9 Line pairs/8 mm $\curvearrowright \Delta y = 345$ µm, red line is fit with error function.

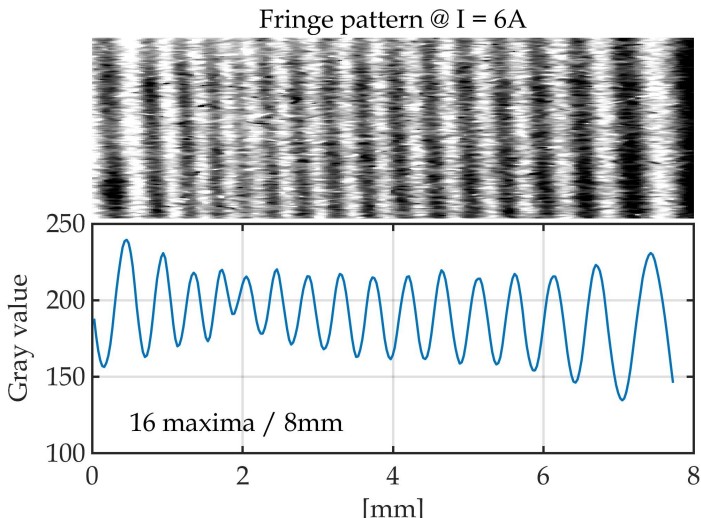

**Figure 13.** Part of Figure 11, image I = 6[A], under-laid by the line plot showing the varying contrast in the image.

### 3.3. Lateral Magnetic Spatial Resolution

The spatial magnetic resolution in the case of polarized neutron imaging was also investigated in the case of lateral spatial resolution. In the coil experiment of Figure 11, the B-field increased and the number of spin rotations (spin flips) along the flight path (y-direction) determined. One measures the number of spin flips (number of fringes) as a function of the B-field for a given distance. So one acquires information about the shortest path length a spin-flip can occur (longitudinal magnetic spatial resolution).

This experiment determines the *largest* B-field that can be measured with a certain spatial instrumental resolution. However, both the largest and the smallest B-field to be detected depend on the wavelength of the neutron. B and thus the path length for a spin-flip are inverse functions to the neutron wavelength, as given by Equation (6) and seen in Figure 2. Due to knowledge of horizontal and vertical divergences $\phi_{hor}$ as described in Section 2.3 the fringe pattern must be adapted to the expected resolution. The detector pixel size (also known) must much smaller than the produced fringe pattern, so that the measured convolution provides useful results.

To determine the lateral spatial resolution, one must measure the spatial resolution with respect to $\Delta x$ and $\Delta z$, varying only the position of a fringe pattern on the 2D detector. This was done by a magnetized quadratic coil (I = 3A) placed on a µm precise stage and moving it up by 50 steps each of one 50 µm, so that the pattern, generated by the

magnetized coil and the spin analyzer, is shifted 2.5 mm above the 2D pixel array of the detector. For each step, an image was taken with polarized neutrons. If the fringe pattern moves perpendicular to the detector (horizontal) lines, the movement of the pattern produces different gray values in the detector lines, which can be plotted as a line graph (Figure 14).

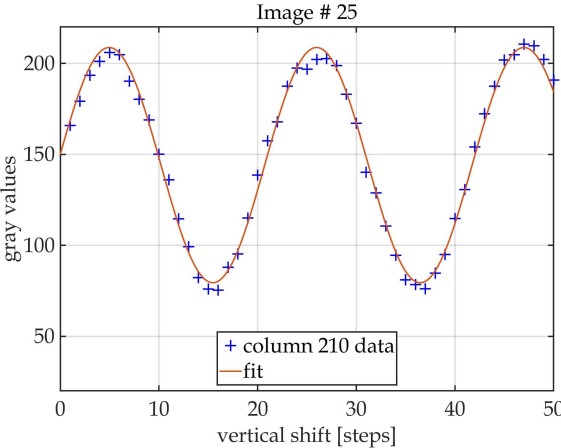

**Figure 14.** Vertical movement of the fringe pattern, each step = 50 μm, total shift = 2.5 mm.

Figure 15 shows five fringe patterns from the radiography image I = 3A, B = 16 mT (Figure 11), four images each shifted by 3 pixel = 81 μm, showing a change from maximum intensity to minimum intensity (spin-flip) for a shift of 12 pixel = 324 μm. A detailed look at different vertical shifts also shows that the distance for a spin-flip varies remarkably. Depending on which part of the image is used, one counts 10 fringes (line pairs, maximum-maximum distance) within 7.6 mm corresponding to $2\Delta z = 0.76$ μm or 380 μm path length for a spin-flip or five fringes within 616 μm or 308 μm for a spin-flip.

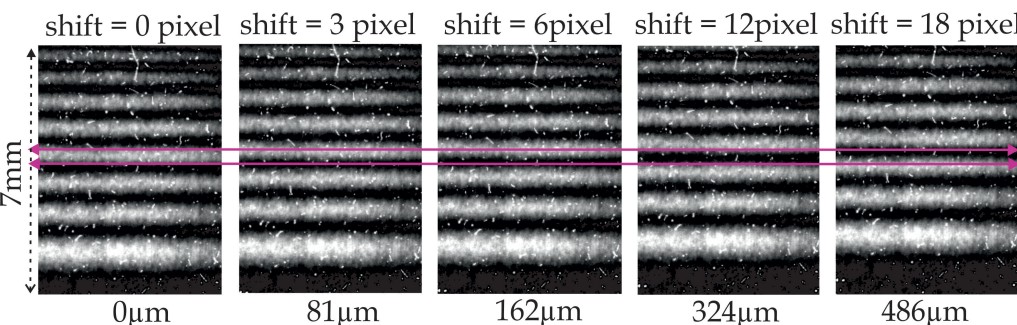

**Figure 15.** Five fringe pattern produced with of quadratic coil (from Figure 11, I = 3.5A) shifted by 3...18 pixel, pattern and fringes rotated by 90°, length of the coil = 150 mm, inner Al block = 11 mm × 11 mm, outer diameter = 35 mm, wire = Cu, ⌀ = 0.90(1) mm, resistance = 1.2 Ω, 614 windings.

If one plots two different shift positions of the fringe pattern, one obtains Figure 16.

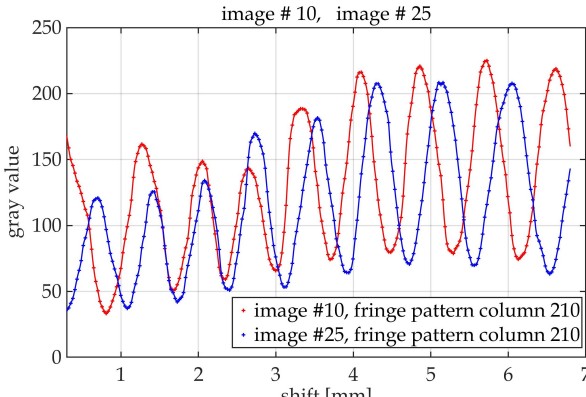

**Figure 16.** Line plot from image 10 and 25, mean distance maximum-maximum = 381(46) μm, see also Table 1.

This was checked in detail with the result that the periodicity of the measured patterns varies from position to position: Table 1 shows mean spin-flip distances for different shift positions. Positions no. 0 and no. 50 show the largest values for a spin-flip and deviations (396(87) μm and 415(73) μm, respectively.

**Table 1.** Fringe pattern.

| Shift Nr. | Spin-Flip Distance [μm] | Standard Dev. [μm] |
|:---:|:---:|:---:|
| 0 | 396 | 87 |
| 10 | 381 | 51 |
| 25 | 381 | 46 |
| 30 | 380 | 48 |
| 50 | 415 | 73 |

The experimental results of the lateral spatial resolution $\Delta z$ (380(48) μm) agree well with the (vertical) geometric spatial resolution (350 μm) and with the calculated result of the convolution of the fringe pattern with the 2D detector (pixel array) (376 μm). Due to the vertical beam divergence $\phi_{vert} = 2\phi_{horiz}$, $\Delta x$ can be given as $\frac{1}{2}\Delta z = 190$ μm.

## 4. Summary

In this paper, at first some basics on spatial resolution and modulation transfer function determination are given and it is proposed to obtain the modulation transfer function from edge image experiments. The edge data are fit to the convolution of a Gauß function with a step function, resulting in a smooth function for the edge image (and for the Gauß function) the Fourier transform is then applied to the (already smoothed) Gauß function, yielding the corresponding modulation transfer function and thus the spatial resolution at 10% contrast.

In this setup (PONTO II) the spatial resolution as a function of the distance of the edge from the detector differs significantly from the pin hole—source geometry. The use of a mosaic monochromator increased the L/D ratio by a factor of 2.2 from L/D(pin hole) = 573 to L/D (mosaic monochromator) = 1270, which has already been demonstrated with another instrument [41]. The spatial resolution for unpolarized neutrons at PONTO II was determined to be 57(3) μm for an object at a distance of 5 mm from the detector, which is an improvement from earlier measurements (80 μm) [35].

When determining spatial resolution in the case of polarized neutron imaging, it is necessary to distinguish longitudinal and lateral resolution. The spin depolarization was measured using a special neutron supermirror device (SwissNeutronics) as polarizer and analyzer elements. The measurement is governed by the Larmor precession $\omega_L$ resp. by the spin rotation angle $\phi$ (see Equation (6)).

The longitudinal resolution was determined by increasing the number of spin flips (fringes) within a given path length. The measured spatial resolution by the modulation transfer function (345(20) μm) agrees well with the calculated value (350 μm) and with the

calculated convolution value (376 μm). The longitudinal resolution refers to the largest B in the sample, the smallest B results from the minimum spin rotation which can still be determined. In addition, the magnetic resolution depends on the incident beam polarization P, and both the largest and smallest B to be detected also depend strongly on the neutron wavelength used. Lateral spatial resolution ($\Delta z$) was measured by moving a given fringe pattern, generated with a magnetized square coil, across the 2D detector in steps of 50 μm. The detector pixels are a known parameter, as is the fringe pattern, so experimental convolution yields the correct values (380(48) μm for a spin-flip. Due to the vertical beam divergence $\phi_{vert} = 2\ \phi_{horiz}$, $\Delta x$ can be given as $\frac{1}{2}\Delta z = 190$ μm.

Finally, it can be stated that the wave nature of neutrons does not affect spatial resolution in magnetic field imaging as described in this paper. The wave nature would cause phase effects, i.e., refraction and diffraction (interference effects). Both interactions require sharp phase transitions [42–46], or ordered structures, for example, in bodies with magnetic domains [54] or in magnetic structures [55], which are not discussed here.

**Author Contributions:** Conceptualization, W.T.; methodology, W.T.; experiments, R.K.; validation, W.T., R.K.; writing—review and editing, W.T., R.K. Both authors have read and agreed to the published version of the manuscript.

**Funding:** This research was funded by the Federal Ministry of Eduction and Research - BMBF, project 05 K10FK1.

**Institutional Review Board Statement:** Not applicable.

**Informed Consent Statement:** Not applicable.

**Data Availability Statement:** Not applicable.

**Acknowledgments:** The authors would like to thank the Helmholtz-Zentrum Berlin für Materialien und Energie (HZB) for strong support of the research work in the framework of the collaboration with the Berlin Hochschule für Technik, University of Applied Sciences. The experiments were performed with the PONTO II instrument at the HZB's BER II research reactor, an instrument funded by the German Federal Ministry of Education and Research (BMBF) and used for radiography and tomography with polarized neutrons. It was installed and operated by the Berlin University of Applied Sciences, Hochschule für Technik, and had to be dismantled at the end of 2019 due to the shutdown of the BER II reactor.

**Conflicts of Interest:** The authors declare no conflict of interest.

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
