# Peer review of "Determination of the Spatial Resolution in the Case of Imaging Magnetic Fields by Polarized Neutrons"

_applsci, doi:10.3390/app11156973_

Round 1
Reviewer 1 Report
This paper considers spatial resolution in neutron imaging with polarized neutrons. The authors propose a method to produce fringes of magnetic field that can be used to probe the resolution of polarized-neutron imaging instruments.
In my opinion, the paper lacks a clear description of the imaging method and the definition of the resolution from the very beginning (in the Introduction), and immediately jumps into technical details that are impossible to follow without understanding what exactly is being imaged. This makes it difficult to follow. It would be very helpful to give a sketch of the experimental setup and name particular neutron instruments to which this theory applies before going into the details. I consider myself an expert in neutron scattering, but not in neutron imaging. To my understanding, the term "neutron imaging" can apply either to physical objects in direct space (e.g. fuel cells, engines, PCBs, etc.) or to distributions of magnetic fields, and the present paper makes the impression that only the magnetic field imaging is considered (as follows from the title). However, the introduction starts with defining the resolution using the "Siemens-star" and absorbing gratings, which clearly cannot be accomplished for magnetic fields, so why mention them at all? At least, it is necessary to explain that this introduction refers to a completely different method of imaging absorption contrast rather than magnetic-field contrast.
The text contains stylistic and typographic mistakes that complicate reading even more. For instance,
1. Abstract: "This resolution resolution refers to..."
2. On top of page 2: "The first one uses a so-called "Siemens-star" and the second one by determining..." --- This is poor English, because the second subject lacks a verb.
3. Bottom of page 2: "...an accuracy of X-ray imaging is obtained" --- what is meant by accuracy here? Shouldn't it be resolution?
4. On top of page 2: "...all have in common a spin analysis system which is part of the resolution function" --- how can a system (device) be part of a function (mathematical object)?
5. Bottom of page 3: "can best represented" -> "can be best represented".
6. Eq. (1): In the conventional definition of the Gaussian function, the factor 2 in the denominator under the exponent should not be squared, it should come in front of the bracket, otherwise sigma is no longer the standard deviation. This function also lacks an amplitude prefactor.
7. Top of page 4: "...and s the theoretical curve line of the edge" -- missing verb.
8. Just before section 2.2: "one can assume of a uniform spatial resolution" -> "one can assume a uniform spatial resolution".
I do not list all other typos, as I think it is the task of the authors to proofread their paper before the submission. This certainly was not done in this case.
The neutron gyromagnetic ratio is given on page 5 with 10 significant digits (is this precision really important?), but it differs already in the 5th digit from the 2019 CODATA value [http://physics.nist.gov/cgi-bin/cuu/Value?gamman] and is given without any reference. It also exceeds the CODATA value by 2 digits in precision. Where was this value taken from? The only place where I found the same number is in the book chapter "Neutron Radiography and Tomography" by Wolfgang Treimer (one of the present authors), where it is also given without any reference. In exactly the same chapter in Table 1, the CODATA value 1.83247172 is given (twice!), which contradicts the one given in the text.
It difficult for me to judge about the correctness of the main part of the text, as it is written in a rather cryptic language full of technicalities and professional slang. While I assume an expert in neutron imaging could understand it better, it is certainly not very clear to an average physicist, even to the one specializing with neutron scattering. Occasionally one also encounters totally arbitrary numbers (e.g. 614 windings in Fig. 5) and is left wondering why this particular value was chosen. Does it correspond to the actual coil geometry in a specific experiment?
My conclusion is that the paper paper contains useful results, but the presentation should be improved. The authors should consider adding a more comprehensive introduction, restructuring and proofreading the text. The current problem is that it is simply very difficult to read.
Author Response
Manuscript ID applsci-1242497
Determination of the spatial resolution in the case of imaging magnetic fields by polarized neutrons
This paper considers spatial resolution in neutron imaging with polarized neutrons. The authors propose a method to produce fringes of magnetic field that can be used to probe the resolution of polarized-neutron imaging instruments.
In my opinion, the paper lacks a clear description of the imaging method and the definition of the resolution from the very beginning (in the Introduction), and immediately jumps into technical details that are impossible to follow without understanding what exactly is being imaged. This makes it difficult to follow. It would be very helpful to give a sketch of the experimental setup and name particular neutron instruments to which this theory applies before going into the details. I consider myself an expert in neutron scattering, but not in neutron imaging. To my understanding, the term "neutron imaging" can apply either to physical objects in direct space (e.g. fuel cells, engines, PCBs, etc.) or to distributions of magnetic fields, and the present paper makes the impression that only the magnetic field imaging is considered (as follows from the title). However, the introduction starts with defining the resolution using the "Siemens-star" and absorbing gratings, which clearly cannot be accomplished for magnetic fields, so why mention them at all? At least, it is necessary to explain that this introduction refers to a completely different method of imaging absorption contrast rather than magnetic-field contrast.
Answers:
The text contains stylistic and typographic mistakes that complicate reading even more. For instance,
- Abstract: "This resolution resolution refers to..."
corrected - On top of page 2: "The first one uses a so-called "Siemens-star" and the second one by determining..." --- This is poor English, because the second subject lacks a verb.
corrected - Bottom of page 2: "...an accuracy of X-ray imaging is obtained" --- what is meant by accuracy here? Shouldn't it be resolution?
yes, - corrected - On top of page 2: "...all have in common a spin analysis system which is part of the resolution function" --- how can a system (device) be part of a function (mathematical object)?
yes, we have corrected: .. but all have in common a spin analysis system which
determines the resolution of the imaging system. - Bottom of page 3: "can best represented" -> "can be best represented".
corrected - Eq. (1): In the conventional definition of the Gaussian function, the factor 2 in the denominator under the exponent should not be squared, it should come in front of the bracket, otherwise sigma is no longer the standard deviation. This function also lacks an amplitude prefactor.
For the determination and fit of the edge the used Gaussian function is most suitable.
Note that one is only interested in obtaining a smooth error function (FT of Gauss)
and hence the MTF. - Top of page 4: "...and s the theoretical curve line of the edge" -- missing verb.
sorry, “ .. “ must be cancelled; corrected: is cancelled. - Just before section 2.2: "one can assume of a uniform spatial resolution" -> "one can assume a uniform spatial resolution".
corrected
I do not list all other typos, as I think it is the task of the authors to proofread their paper before the submission. This certainly was not done in this case.
Anwer:
We carefully read and corrected the MS
The neutron gyromagnetic ratio is given on page 5 with 10 significant digits (is this precision really important?), but it differs already in the 5th digit from the 2019 CODATA value [http://physics.nist.gov/cgi-bin/cuu/Value?gamman] and is given without any reference. It also exceeds the CODATA value by 2 digits in precision. Where was this value taken from? The only place where I found the same number is in the book chapter "Neutron Radiography and Tomography" by Wolfgang Treimer (one of the present authors), where it is also given without any reference. In exactly the same chapter in Table 1, the CODATA value 1.83247172 is given (twice!), which contradicts the one given in the text.
Answer:
We (I) deeply regret this misprint. It does not make any sense to use more than four digits for our calculations, in all cases you mentioned we used the data from NIST: http://physics.nist.gov/cgi-bin/cuu/Value?gamman 1.83247172x10^8s^-1T^-1.
I wonder how this 10 digit number came in the MS. I suppose – after all – a misprint. In the book chapter "Neutron Radiography and Tomography," I cite NIST with references 15 and 16, from which I have taken the numbers in Table 1.
We have corrected the value and added NIST CODATA reference.
It difficult for me to judge about the correctness of the main part of the text, as it is written in a rather cryptic language full of technicalities and professional slang. While I assume an expert in neutron imaging could understand it better, it is certainly not very clear to an average physicist, even to the one specializing with neutron scattering.
Answer:
I agree that this paper is quite “mathematically written”, and we tried to write a paper dedicated to people which perform imaging with polarized neutrons and to omit technicalities and professional slang.
Up to now I found no publication which deals with this subject.
Occasionally one also encounters totally arbitrary numbers (e.g. 614 windings in Fig. 5) and is left wondering why this particular value was chosen. Does it correspond to the actual coil geometry in a specific experiment?
Answer:
We have ordered coils of a certain size that are suitable for our experiments and that
can produce magnetic fields of about 30mT. The number of 614 turns is simply the
number we got from the manufacturing company.
My conclusion is that the paper contains useful results, but the presentation should be improved.
The authors should consider adding a more comprehensive introduction, restructuring and proofreading the text. The current problem is that it is simply very difficult to read.
Answer:
We agree with the Reviewer and have added more explanatory text in the introduction and tried to add explanatory text .
Acknowledgment
We thank the Reviewer for his careful reading, for corrections of typographical and numerical errors, and for constructive suggestions to improve the style and understanding of the work.
We have considered all of his suggestions to the best of our ability and hope to have answered and clarified all questions to the reviewer's satisfaction.

Reviewer 2 Report
The authors report on the determination of the spatial resolution in the case of imaging magnetic fields by polarized neutrons. Comments are provided below:
Page 1
- line 21: parameters, not parameter.
- line 22-23: confusing sentence. Please reconsider it.
- line 23: should instead of must.
- line 32: deleted one of "resolution".
Page 2
- line 24: two "but"; please reconsider the sentence.
- line 32: why (or what limits) the spatial detector resolution to the 1-micron size?
- line 39-42: the sentence is confusing; what does it mean "however, today already an accuracy of X-ray imaging is obtained"?
Page 3
- line 5: should "special" be "spatial"?
- line 18-22: confusing sentence given the "however" (2x) included. Please revise.
Page 4
- Please add reference to Shannon theorem.
- Equation 4: delta apparently appears for the first time, but is not defined.
- line 14: It is not clear why "The frequency at 10% contrast gives the spatial resolution in the image." Please explain.
Page 6:
- Figure 4: it is not clear the relevance/meaning of it for the article.
- Equation 8: please add a reference.
Page 7
- Figure 5: is it experimental or calculated? Please indicate.
- Line 4: please delete one of the "coil" words.
Page 14
- Line 17: which device provides a "nm precise lifting"? Mechanical systems operate with the precision of microns.
Author Response
ANSWER TO REVIEWER II
The authors report on a study of magnetic field trapping in niobium, and about how trapping may be affected by small AC magnetic fields. The polarized neutron study on Nb single crystals produced quantitative and spatially resolved magnetic flux values which were compared to calculated values. The hypothesis of alternating fields affecting the trapping behaviour remained inconclusive but could not be ruled out for an untreated Nb sample. Overall, the data are sound and the conclusions are reasonable. The study is an excellent example of how polarised neutron imaging is unique for visualizing flux trapping.
The authors were careful not to over-interpret their results: slight frequency dependencies visible in the recorded images (Fig. 3 - Fig. 6) and in the analysed images (Fig. 10) were not considered significant; a trapped field decreases with frequencies for the treated sample.
The manuscript needs minor corrections and adjustments. I recommend that the following points are addressed:
Page 2 Line 33: The text mentions spin analysers before the experiment is described. Also, the stray field was mentioned (0.19mT) at a sample-analyser distance of 18cm etc. Such details seem to be misplaced in the theory section. I wonder if such details can be moved to the experimental section.
Answer:
We agree and moved this part at the end of section III. A, where it fits much better.
Page 3 Fig 1: The label says: Pb sample. This needs to be: Nb sample
Yes, we have changed the label in Fig.1
Page 5 Line 3: The central pixel areas were selected to determine the trapped field etc. It would be good if the authors gave some more detail about which region of interest was selected. Was it one row, or several rows? Is this related to the yellow box in Fig. 4?
Answer:
Yes, we clarify and add (please see also text):
The size of each image was 500 x 600 pixel (21.5mm x 25.8mm), the area of integration was 1024 pixel = 32 pixel x 32 pixel (1.37 mm x 1.376mm = 1.893mm2), the yellow boxes in Fig.4 show the projected volumes of yellow circles in Fig.6.
Page 6 Line 53: There is some inconsistency with regard to the sample crystallinity: … polycrystalline sample … On page 3 line 1 the samples are introduced as single crystalline.
Answer:
The used Nb sample was polycrystalline,
The manuscript is well written overall. There are a number of formatting and language issues to be addressed:
- several references appear non-sequentially on pages 1+2: #18-#23; #23; #26; #28
The authors appended enlarged figures 3-8 at the end of the manuscript. This helped reading the paper a lot. One should think about enlarging those same figures in the manuscript.
OK that must be done be the journal - Leave a blank between values and units (multiple occasions in the manuscript).
OK, done
Answer:
Concerning non-sequentially: The references are given corresponding to the text, and authors are cited belonging to different subjects, however we agree with Reviewer
Language:
- Page 1 Line 40: what is RRR?
Is added to text: residual resistivity ratio - Page 1 Line 41: which methods?
Methods cited in [1] –[11] - Page 1 Line 87: led (instead of lead) :
Corrected - Page 2 line 80 ( and at several other places): flux trap; amount of trapping; etc : maybe better: flux trapping; level of trapping; etc
Throughout the paper flux trapping is used, - At several places: should it be: buffered chemical polished, instead of: buffered chemically polished?
buffered chemically polished is correct - Page 2 line 3: complementary (instead of: complimentary
Corrected - Page 2 line 45 only (remove preceding comma); appears more than once in the paper;
Corrected - Page 2 line 60 … alpha is constant in this calculation …
- Corrected
- Page 2 line 82 what is being quantified?
Amount/level of flux trapping - Page 3 line 13 … were …
OK - Page 3 line 55/56 images? items?
corrected: images - Page 4 line 22 fourth row in Fig. 4 is referred to but there are only three rows in the figure
there are only three rows …. is corrected - Page 4 line 29 … for the sample orientation (instead of: in the sample orientation); occurs more than once in the paper;
Through the paper corrected (in / for) - Page 5 Fig 7 caption: (… first row in Fig. 3)
corrected - Page 5 line 30: weaker (instead of smaller)
corrected - Page 6 line 32: remove ‘themselves’
removed - Page 6 line 4: there is no fourth row in Fig. 6
corrected - Page 7 line 30 small angle scattering (instead of: angled)
corrected - Page 7 line 62 … assess … (instead of: access)
corrected
We would like to thank the reviewer for all the errors found, precise comments and suggestions that really helped to improve our work. We have addressed all points in the text and captions and hope to have answered all questions satisfactorily.
Many thanks
W. Treimer

Reviewer 3 Report
The authors convincingly prove that they master the presented method of neutron imaging.
The subject is well explained. The article contains detailed information.
Some typos and repetitions need to be corrected. Since the overall quality of the text is high, this will be easy to do. Authors just need to carefully re-read the entire text of the article one more time.
The description of neutron detectors of high spatial resolution and the list of references do not correspond to each other precisely. Maybe you can add [J. Jakubek et al, Nucl. Instr. Meth. A 560 (2006) 143] for Medipix detectors and [S. Baessler et al, Compt. Rend. Phys. 12 (2011) 729] for the mentioned "observing tracks from neutron capture events".
Neutron analyzers and polarizers are an important part of the method described in the article, but they themselves are not even mentioned. I propose to add a paragraph describing the polarizers and analyzers used and give some related references.
This manuscript can be published after minor revisions.
Author Response
REVIEWER
The authors convincingly prove that they master the presented method of neutron imaging.
The subject is well explained. The article contains detailed information.
Some typos and repetitions need to be corrected. Since the overall quality of the text is high, this will be easy to do. Authors just need to carefully re-read the entire text of the article one more time.
Answer:
We have corrected grammar and spelling errors to the best of our knowledge.
The description of neutron detectors of high spatial resolution and the list of references do not correspond to each other precisely. Maybe you can add [J. Jakubek et al, Nucl. Instr. Meth. A 560 (2006) 143] for Medipix detectors and [S. Baessler et al, Compt. Rend. Phys. 12 (2011) 729] for the mentioned "observing tracks from neutron capture events".
Thank you for this note, we have added both references.
Neutron analyzers and polarizers are an important part of the method described in the article, but they themselves are not even mentioned. I propose to add a paragraph describing the polarizers and analyzers used and give some related references.
We agree, we have added details of neutron polarizer (and analyzer)
This manuscript can be published after minor revisions.
Acknowledgment
We thank the Reviewer for his careful reading, for corrections of typographical and numerical errors, and for constructive suggestions to improve the style and understanding of the work.
We have considered all of his suggestions to the best of our ability and hope to have answered and clarified all questions to the reviewer's satisfaction.

Reviewer 4 Report
This manuscript reports the way how the spatial resolution of the polarized-neutron imaging of magnetic field distribution. This work is based on the established semi-classical equations of motion of neutrons, i.e., geometrical optics and Larmor precession. The authors applied the method to real imaging experiments and successfully estimated the longitudinal and lateral spatial resolutions of the imaging. While I think that the work is worth publishing in Applied Sciences, I would like to request the authors to append how the wave nature of neutrons affects the spatial resolution of the magnetic-field imaging.
Author Response
REVIEWER
This manuscript reports the way how the spatial resolution of the polarized-neutron imaging of magnetic field distribution. This work is based on the established semi-classical equations of motion of neutrons, i.e., geometrical optics and Larmor precession. The authors applied the method to real imaging experiments and successfully estimated the longitudinal and lateral spatial resolutions of the imaging. While I think that the work is worth publishing in Applied Sciences, I would like to request the authors to append how the wave nature of neutrons affects the spatial resolution of the magnetic-field imaging.
Answer to the Reviewer:
In order to answer this interesting comment of the Reviewer we added to the conclusions:
“Finally it can be stated that the wave nature of neutrons does not affect spatial resolution in magnetic field imaging as described in this paper. The wave nature would cause phase effects, i.e. refraction and diffraction (interference effects). Both interactions require sharp phase transitions \cite{TreimerPLA2002, Strobl2008NIM, Tremsin2012, StroblAPL2007, Treimer2003}, or ordered structures, for example, in bodies with magnetic domains \cite{HubertSchaefer} or in magnetic structures \cite{Chatterji2006}, which are not discussed here.”
We thank the Reviewer for his careful reading, we have considered his suggestion and added text to the best of our ability and hope to have answered and clarified his question to the reviewer's satisfaction.
Reviewer 5 Report
I believe that Sec. 2 should be omitted and replaced with a clear description of the experiment. This section contains many mistakes and inaccuracies, and the results are not explicitly used in describing the experimental data. Namely, the authors should
- write what values are measured and what signals are subjected to mathematical processing;
- clearly define which problems can be solved in experiments with unpolarized neutrons, and which only with polarized neutrons;
- why are two different types of experiments presented in the same article?
The authors can present the experimental material and the proposed new methodology for describing the experimental data, if any, in a new paper.
Author Response
REVIEWER
Comments:
I believe that Sec. 2 should be omitted and replaced with a clear description of the experiment.
ANSWER: Please read Chapter 3 and Summary, see answers below.
This section contains many mistakes and inaccuracies, and the results are not explicitly used in describing the experimental data.
ANSWER: There were some typing errors, which were corrected, but results and
experimental data are given -see below.
Namely, the authors should
- write what values are measured and what signals are subjected to mathematical processing;
ANSWER
This MS describes how the spatial resolution is determined for unpolarized and polarized neutrons. Measured values are given in Fig.9 (unpolarized neutrons), Fig.13 for polarized neutrons (longitudinal resolution) and in Fig.16 and Tab.1 for the lateral spatial resolution. Also, read Chapter 3 and Summary. - clearly define which problems can be solved in experiments with unpolarized neutrons, and which only with polarized neutrons;
ANSWER
These questions you ask are not expected to be answered by this MS and are also not the subject of this MS. For this question, read classical neutron books. - why are two different types of experiments presented in the same article?
ANSWER
One has to distinguish once the spatial resolution in imaging for unpolarized and polarized neutrons, for polarized neutrons one has to distinguish between longitudinal and transverse spatial resolution (read Chapter 2).
The authors can present the experimental material and the proposed new methodology for describing the experimental data, if any, in a new paper.
ANSWER
Section 2 presents the mathematics necessary to understand the experiments for imaging with unpolarized neutrons and then with polarized neutrons (longitudinal and lateral spatial resolution). Omitting this section therefor makes no sense. In this MS all methods, mathematics and based on them all experiments were described in detail, the results were given in detail and discussed. We see no reason to write a new paper on this subject, also because of the large overall agreement with this MS by all other reviewers.

Round 2
Reviewer 1 Report
I have no additional suggestions for improvements. The authors seem to have taken my remarks into account. The paper still remains very technical and accessible only to experts, but according to the authors that is what it is supposed to be. I believe the paper can be published in its present form.
Reviewer 2 Report
The authors addressed my comments.
Reviewer 5 Report
The authors did not answer any questions, the text of the article has not undergone significant changes.
Sec. 2 contains naive scientific results, the equations contain many typos.